# Predicting Group Contribution Behaviour in a Public Goods Game from Face-to-Face Communication

**DOI:** 10.3390/s19122786

**Published:** 2019-06-21

**Authors:** Ehsan Othman, Frerk Saxen, Dmitri Bershadskyy, Philipp Werner, Ayoub Al-Hamadi, Joachim Weimann

**Affiliations:** 1Neuro-Information Technology group, Institute for Information Technology and Communications, Otto-von-Guericke University Magdeburg, 39106 Magdeburg, Germany; Frerk.Saxen@ovgu.de (F.S.); Philipp.Werner@ovgu.de (P.W.); Ayoub.Al-Hamadi@ovgu.de (A.A.-H.); 2Halle Institute for Economic Research, 06108 Halle, Germany; Dmitri.Bershadskyy@iwh-halle.de; 3Faculty of Economics and Management, Otto-von-Guericke University Magdeburg, 39106 Magdeburg, Germany; Joachim.Weimann@ovgu.de

**Keywords:** face-to-face communications (FFC), voluntary contribution mechanism (VCM), random forest classification (RFc), facial activity descriptors (FADs), group activity descriptors (GADs), public goods game

## Abstract

Experimental economic laboratories run many studies to test theoretical predictions with actual human behaviour, including public goods games. With this experiment, participants in a group have the option to invest money in a public account or to keep it. All the invested money is multiplied and then evenly distributed. This structure incentivizes free riding, resulting in contributions to the public goods declining over time. Face-to-face Communication (FFC) diminishes free riding and thus positively affects contribution behaviour, but the question of how has remained mostly unknown. In this paper, we investigate two communication channels, aiming to explain what promotes cooperation and discourages free riding. Firstly, the facial expressions of the group in the 3-minute FFC videos are automatically analysed to predict the group behaviour towards the end of the game. The proposed automatic facial expressions analysis approach uses a new group activity descriptor and utilises random forest classification. Secondly, the contents of FFC are investigated by categorising strategy-relevant topics and using meta-data. The results show that it is possible to predict whether the group will fully contribute to the end of the games based on facial expression data from three minutes of FFC, but deeper understanding requires a larger dataset. Facial expression analysis and content analysis found that FFC and talking until the very end had a significant, positive effect on the contributions.

## 1. Introduction

Cooperation is one of the major foundations of economic actions. Given its essential role, a lot of research in economics focuses on how cooperation can be improved. At the heart of this research is the evidence that communication is a highly effective tool to mediate social dilemmas [1]. A very straightforward way to analyse social dilemmas in experimental economics research is to use the public goods game, an experimental setup that allows for cooperation and incentivises free riding. However, the main research goal is to sustain cooperation and stop free riding [2]. An especially non-intrusive and simultaneously very effective way of doing this is to provide the participants of the experiment with an opportunity to communicate with each other prior to the public goods game itself. This free-form, face-to-face communication (FFC) induces group members to make more socially optimal contributions in public goods games independent of whether the FFC takes place in person or in a video-conference [3]. In this context, we introduce the following issue to the computer vision and behaviour understanding domain: the prediction and understanding of (future) behaviour based on FFC analysis. We automatically analyse the group’s FFC in a laboratory public goods experiment with the goal of explaining if (and how) non-verbal communication enables the prediction of the group’s future behaviour.

In this paper, we predict the end-game contribution behaviour of groups in a public goods game [4] after a 3-minute video-based communication using automatic facial expression analysis. The models that predict the group behaviour in the last phase of the games can help to identify groups that are going to provide socially sub-optimal contribution rates to the public good prior to their contributions. The ultimate goal could therefore be to intervene (e.g., nudges, efficiency providing formal institutions) only when the prediction based on a priori available information concludes that the group needs another push towards the social optimum. Given the specific structure of this financially incentivised laboratory experiment, the subjects act as free riders essentially only in the final stage of the experiment. The underlying dilemma of the public goods games is that socially and individually optimal behaviour do not coincide. The discrepancy becomes more apparent towards the end of the game when the benefits of cooperation decrease. In the last stage of the game there is no future cooperation, thus individuals in experiments without communication by majority maximise own payoffs by free-riding. Communication increases contribution rates on average, yet does not dispose the end-game effect completely [3,5]. We train a binary classifier that predicts whether the group will contribute fully in the very last period or defect. In this public goods experiment, each group had four participants; we assume that the end-game behaviour of each individual is influenced by the prior contributions of the other participants within his/her group. Therefore, we do not predict the contribution of each participant but of the entire group. The size of the dataset (described in Section 3) is small: it consists of 24 sessions including in total 127 different groups. The same subject might appear in several groups, but only in one session. For smaller datasets, deep networks tend to overfit. As a result, classical machine learning approaches often outperform deep networks for small datasets. That is why we use a classical approach here. To train person-independent models using classical machine learning approaches, we perform leave-one-session-out cross-validation. This ensures that the subjects do not appear in the training and test set simultaneously.

For the independently conducted content analysis, verbal conversations were first transcribed into texts. Subsequently, the texts were classified using binary content parameters, e.g., whether a specific topic [3] was raised (1) or not (0), and meta parameters of the conversations, e.g., individual and group word counts. The obtained variables were analysed with respect to the contribution behaviour and the collected demographic information.

The remainder of this paper is organised as follows. Section 1.1 describes the related works to address the relationship between this paper and the reference papers, while Section 1.2 describes the contribution of this work. Section 2 outlines the proposed automatic facial expression analysis approach. Section 3 describes the data and the experimental setup that was used to collect the data. Section 4 presents the experiments that were carried out, and their results. It also provides the results of the content analysis. We conclude the paper with a discussion in Section 5.

### 1.1. Related Work

Studies have shown that a large part of affective communication takes place either nonverbally or paralinguistically through audio or visual signals [6,7]. Besides the verbal information of opinions and desires, human interaction also involves nonverbal visual cues (through gestures, body posture, facial expressions and gaze) [8,9] which are widely and often unconsciously used in human communication [10,11]. Facial expressions are more than a static appearance cue with specific physical features. They are important signals of emotional states [12], communicate our intentions to others (as communication is often carried out face-to-face) [9,13,14,15], and aid in predicting human behaviour [16,17,18]. Facial expressions play an important role in interpersonal communication, and interpersonal behaviour of individuals is influenced by social context [19,20]. Interpersonal communication while displaying positive facial expressions increases individual performance in groups and contributes immensely to increasing, for example, workforce performance and overall organisational productivity [21]. In general, interpersonal communication has been proven to be very useful in different social groups. Facial expressions are used as an effective tool in behavioural research and widely used in cognitive science and social interactions.

Automatic analysis of social behaviour, in particular of face-to-face group conversations, is an emerging field of research in several domains such as human computer interaction, machine learning and computer vision [16,22,23]. The ultimate aim is to infer human behaviour by observing and automatically analysing the interaction of the group conversation taken from the audio and video channels. Jayagopi et al. (2009) describe a systematic study that investigates the characterisation of dominant behaviour in group conversations from fully automatic nonverbal (audio and visual) activity cues [24]. Jaques et al. (2016) present how a machine learning classifier can be trained using the facial expressions of one-minute segments of the conversation to predict whether a participant will experience bonding up to twenty minutes later [25]. Many automatic facial expressions were recognised on the basis of the display of individual facial Action Units (AUs) and their combinations. AUs are defined by the facial action coding system of Ekman and Friesen [26], which is the most commonly used method for coding facial expressions. Previous studies proved that combinations of AUs can account for more variation in behaviour than single AUs alone [27]. In addition, several lines of evidence suggest that the combination of AUs predict behaviour more precisely [28].

Research has shown that group behaviour in the first few minutes (3 min) is highly correlated with the decisions made and actions taken in the future [29,30,31]. Thus, using machine learning coupled with computer vision allows computers to predict future behaviours. Recent research [32,33,34,35] shows how to automatically recognise emotions from videos or still images of individuals. However, group behaviour is barely addressed in the literature and studies do not deal with interpersonal communication and its impact on future decisions. Furthermore, at present, the typical publicly available facial expression databases contain recordings of spontaneous facial expressions and corresponding FACS annotations of one individual. But only one available database, the Sayette Group Formation Task (GFT) database, includes multiple interacting participants, despite the important role that facial expressions play in interpersonal communication. In GFT, there are three subjects in the videos interacting naturally and the purpose of this database is to estimate the overall emotion of the group interaction in order to study social interaction [36]. The originality of GFT is to have three subjects instead of two as in RECOLA [37]. However, with GFT, the future prediction of facial expressions for group participants is a difficult task, mainly because there are no direct decisions. Furthermore, there is no financial incentive involved in GFT to force decisions, so it cannot be utilised to study financial interactions. Ten Brinke et al. (2016) and Bonnefon et al. (2017) discuss the potential positive effects of incentivising experiments to detect deception or cooperation [38,39]. Therefore, in this paper, we utilise a special database from a laboratory public goods experiment of [40] that provides binary financial decisions after three minutes of FFC, as described in Section 2. Our prior research on the effect of communication on the voluntary contribution to public goods has clearly shown that communication for three minutes strongly increases the ability to cooperate, especially if the communication happens to be face-to-face [5]. This finding strongly supports the hypothesis that facial expressions play a major role in group communication processes.

Some researchers implement “human assessment” methods to predict the behaviour of subjects in laboratory experiments. In such cases, individuals attempt to guess whether other people will cooperate or not in setups similar to ours [41,42]. This paper is the first that investigates the facial expressions of a group in videos to predict their behaviour in a public goods experiment using facial expression recognition software. With respect to possible applications of public goods experiments, we refer to the prominent literature review by Chaudhuri [2]. In the broad sense, the results of such laboratory experiments are applied to, for example, charitable giving, managing natural resources, tax compliance and work relations. In a more narrow sense, the availability of tools automatically detecting free riding can aid (1) public transport systems that suffer losses due to fare evasion [43,44]; or (2) organisations that are interested in knowing whether a team will work together well, which has implications for group productivity [45].

### 1.2. Contribution

The main contribution of this paper is twofold. First, we introduce the application of computer vision and machine learning to experimental economics research in order to analyse group behaviour. Second, we introduce the economics topic of free riding in groups to the behaviour analysis community. Previous research has investigated the impact of communication on cooperation behaviour in groups [46]. Yet, so far, there has been no application that uses facial expressions to predict group behaviour after face-to-face communication (FFC) of free discussion. Thus, to the best of our knowledge, this is the first work that investigates the facial expressions of a group in videos to predict their behaviour in public goods experiment using facial expression recognition software. In this context, we propose an automatic facial expression analysis approach for the prediction of future group behaviour from pre-play FFC videos, which includes a novel group activity descriptor. Using the proposed approach, we show that a machine learning classifier can be trained to predict whether the group will fully contribute to the end of the games based on facial expression data from three minutes of communication. The spoken contents of the FFC video are investigated by categorising words in topics and using meta-data.

## 2. Automatic Facial Expression Analysis Approach

In the setup of a public goods game, all the group members benefit from the public good, regardless of their own contribution [4]. Within the scope of this study, the participants communicate with each other prior to secretly choosing how much of their financial endowment they want to put into the public pot. Both facial expressions and content analysis are conducted independently to investigate whether the FFC induces the group members to sustain cooperation and stop free riding. Figure 1 shows an overview of the automatic facial expression analysis for the FFC video to predict the end-game behaviour of the groups. Each group in the FFC video has four participants (subjects). We do not predict the contribution of each subject but of the entire group. First, we use OpenFace [47] to extract head pose and AUs features (facial features, FF) from each frame for each participant (see Section 2.1). Second, the FF are condensed into descriptors [48] for each individual face from all the frames (see Section 2.2). Third, the descriptors for each of the 4 individuals are concatenated in all possible ways to form the group activity descriptors and all of them receive the same label (see Section 2.3). Finally, we classify the group activity descriptors individually and the outcomes are evaluated and averaged to calculate the classification performance (see Section 2.4).

### 2.1. Facial Features (FF)

The first step in our analysis pipeline uses OpenFace [47] to extract the facial features from each individual face and frame. OpenFace detects the face, facial landmarks, extracts Action Units (AUs), and estimates head pose (see Figure 1). The frame-based AUs and head pose form the facial features. To analyse the importance of the features we group them into four subsets as follows. FF1: the first category includes all features (head position, AU presence, and AU intensity); FF2: includes head pose and AU presence features (HP & AUP); FF3: includes head poses and AU intensity features (HP & AUI); FF4: includes manually selected features that OpenFace can estimate robustly based on the study of [47] (see Table 1).

### 2.2. Facial Activity Descriptors (FADs)

For each participant and video, a facial activity descriptor is extracted from the corresponding facial features, as described in Section 2.2 of [49]. In brief, the FF (which differ in length) are condensed in descriptors [48]: First, we smooth each FF with a Butterworth filter (first order, cutoff 1 Hz); Second, we calculate the first and second derivative of the smoothed signal [49]; Third, we extract 16 statistics from the smoothed signals and both derivatives for each facial feature (among others: mean, maximum, standard deviation, time of maximum value and duration in which the time series values are above their mean). In most experiments, we calculate the FAD for each individual from all the frames of the FFC video. However, to analyse the usefulness of specific video parts for the classification, we also extract FADs from temporal segments (as detailed in Section 4).

### 2.3. Group Activity Descriptors (GADs)

The GAD is used to predict group behaviour of the FFC. After we calculate the four FADs of each group, we concatenate them in all 4! = 24 permutations (e.g., 1234, 1243, 1432, …) to form the GADs. This way, each group conversation is represented by 24 instances and each is given the group label. This obviously increases the dataset size, which is favourable because the dataset with 127 FFC videos is quite small. The feature space dimensionality of each GAD for FF1 is 7,296, for FF2 is 4,032, for FF3 is 3,840 and for FF4 is 4,032.

### 2.4. Classification

The (127 × 24 = 3048) GADs are classified individually. The classifier uses leave-one-session-out (24-session) cross-validation, i.e., a subject never appears in a training and test set simultaneously. Cross-validation is a very useful technique for assessing the effectiveness of our models to mitigate overfitting. We hold out the GADs of one session for testing while the rest form the training set. We train the model on the training set and evaluate it on the test set. The cross-validation process is then repeated 24 times, with each of the sessions used exactly once as the test set. In total, 24 sessions provide 24 training sets and 24 test sets. The results of test sets are averaged to form the final performance.

## 3. Dataset

This section introduces the parts of a larger experiment that are relevant to this paper. The collected data and the experimental setup are detailed in [5,40]. The experimental design was executed in z-Tree [50]. The experiment was organised and recruited with the software hroot [51] in the Magdeburg Experimental Laboratory of Economic Research (MaXLab). The participants were students from the Otto-von-Guericke University Magdeburg. The entire experiment consisted of three blocks. The second and third blocks, where the participants had the chance to communicate face-to-face via audio-video communication software prior to the voluntary contribution mechanism (VCM), were analysed. During the communication period of three minutes, the participants were free to discuss anything. The dataset consisted of 127 different groups divided into 24 sessions, with each group having 4 participants (subjects). It was possible for the same subject to appear in several groups, but only in the same session. For the evaluation, we suggest leave-one-session-out cross validation, which avoids subject overlap between the training and the test set and approximates generalisation performance with unseen subjects. For this purpose, the 24 sessions of our dataset were divided into two subsets (training and test set) 24 times: each time a different session formed the test set and the rest formed the training set. Behaviour labels (whether the group contributes fully or not) were provided for both training and test sets, while training set labels were used to learn the classification model and test set labels are used to measure the performance of a classification.

## 4. Experiments and Results

This section provides the experiments and results of the group’s contributions in the public goods games after a group conversation. Section 4.1 describes the setup of experiments and the results. Ablation experiments are described in Section 4.2. Section 4.3 shows the feature importance of FF4, and finally, Section 4.4 discusses the content analysis experiment.

### 4.1. Experimental Setup and Results

We conducted several experiments in order to gain insights into the contribution behaviour classification and to improve the classification accuracy. For reference, we calculate a trivial classier which always votes for the majority class of the training set. Note that this is also done within the leave-on-session-out cross-validation, providing 24 different results (depending on the class distributions of the test sets). This informed guess allows us to have a meaningful comparison with our models.

#### 4.1.1. Model and Feature Selection

The results in Figure 2a show that RFc performs much better than Support Vector Machine (SVM) models. This is consistent to the results of [52] in which classification rates with RFc were significantly better than with SVM. The SVM models perform worse than trivial. This suggests that a large number of features contain noise, which significantly harms the SVM performance (while RFc is quite robust against noise). This also seems to hold true for the different sets of facial features. The selected features (FF4) probably contain less noise than others and thus perform best using the SVM. However, RFc performs better using all features (FF1) and using the selected feature subset (FF4). Therefore, we continue using RFc with FF1 and FF4.

#### 4.1.2. RFc Trees

We investigate the influence of the number of trees on performance and training time. We train 100, 1k, and 5k trees both on FF1 and FF4. The results in Figure 2b show that the accuracy for FF1 and FF4 increases with the number of trees. However, the training time also increases. The runtime in minutes for the entire 24-session cross-validation procedure of training and test sets was plotted to show the factor of runtime increase when the parameters increase. Since the performances gain plateaus at 5k trees, we stopped using more. Models that use FF4 with 1k and 5k trees are better than the models that use FF1 because RFc is more effective when using a small subset of high-dimensional features. Therefore, we continue using RFc (5k trees) with FF4.

#### 4.1.3. Optimal Threshold

In the previous experiments, the RFc is trained on all the GADs of the training set. Each group conversation is represented by 4! = 24 samples and each is given the same label (see Section 2.3). However, the RFc is not aware of this fact and might predict samples of the same group differently. We average the binary prediction outcome of all (4!) samples of the same group and threshold this average value to get the prediction result for that particular group. We determine an optimal threshold that maximises accuracy on the training set.

We calculate the optimal threshold and estimate the final performance by using nested cross-validation. The hyper-parameter search is performed with inner cross-validation, while outer cross-validation computes an unbiased estimate of the expected accuracy of the RFc with the optimised threshold. For each fold of the outer cross validation, the inner cross-validation runs on the outer cross-validation’s training set, trains 23 RFc models and tests them with a predefined set of thresholds. The threshold that yields best accuracy (mean across the 23 inner test sets) is selected as the optimal threshold. In outer cross-validation, we apply RFc with the optimal threshold to the remaining session (1/24) of the data (test set) and calculate the accuracy. In total, we get 24 results, which are averaged to form the final performance. This double cross-validation procedure provides an unbiased estimate of the accuracy for the RFc.

Applying the optimal threshold increases the accuracy of the RFc (5k trees) model (using FF4) from 67.39% to 68.11%. Therefore, we continue using RFc (5k trees) with FF4 and optimal threshold.

#### 4.1.4. Temporal Splits

Computing the FADs for the entire video (see Section 2.2) might reduce the possibility of obtaining time-related features. Because the FADs primarily contain statistical features, it is possible that changes in the signals over time are not captured well enough. Instead of computing the FADs for the entire video, we divide each FFC video into 3, 4 and 5 equally long videos (splits). We extract the FADs and GADs from FF4 of these 12 different splits (3 + 4 + 5 = 12) and combine the GADs in multiple combinations. We train a model for each split and multiple combinations (33 in total, Table 2 shows the full list of splits). In this way, we can capture more time-related features from FFC. We introduce three different categories containing models from different split combinations (beginning models, end models and a combination of beginning and end models). This way we can investigate if there are some parts of the FFC video that are more useful than others.

Figure 3 shows the comparison of accuracy between the uninformed guess (50%), the informed guess (trivial) and three categories of models (combined, beginning and end models). These models are trained using RFc (5k trees) with FF4 and use optimal threshold. The end models perform best in terms of classification performance compared to the uninformed guess, trivial, combined, and beginning models. The end models probably contain more informative features than others. On average, end models predict about 70.17% of the decisions correctly, which is significantly more than guessing, while the combined models predict about 68.57%, the beginning models predict about 68.40% and the trivial models predict about 64.47%. Table 3 shows the results of end models are the best based on p-values, which were obtained through a paired U-test. The results are significant compared with the uniformed guess and trivial models. Other differences failed to reach the significance level, but overall the results are promising for further research.

### 4.2. Ablation Studies

To gain further insights into the improvements obtained by the proposed approach, we summarise the experiments for ablation studies as listed in Table 4, where we aim to examine the effectiveness and contributions of different strategies used in the proposed approach.

Table 4 provides an overview of a series of improvements in RFc models that use the FF4 after applying a series of experiments. The average accuracy is increased from about 65% to 70%. In experiment 1, in which we use all the frames in the FFC video to train RFc (30 trees) with FF4, the models correctly predict 65.29% of decisions, which is slightly better than the informed guess (64.47%). In experiment 2, we use the same setting of experiment 1 except that we increase the number of RFc trees to 5k to increase accuracy. Additionally, the optimal threshold is applied in experiment 3 and prediction accuracy is increased to about 68.11%. In experiment 4, we train RFc (5k trees) using FF4, which are extracted from different splits and combinations in the FFC video to capture the differences between different splits (by obtaining more time-related features) with the optimal threshold. We train RFc (5k trees) with 11 different splits of the combined models (see Table 2). The average accuracy of combined models is 68.57%, while the accuracy of the all frames model is about 68.11%. These results indicate that some splits of the FFC video are not useful for classification. In experiment 5, we investigate whether the first or the last part of the FFC video is most informative. We train RFc (5k trees) with 22 different splits (see Table 2) belonging to the first and the last parts of the FFC video. The beginning models predict about 68%, while the end models predict about 70%, which is significantly more than guessing. However, the accuracies of the end models are not significantly higher than the accuracies of the beginning models.

### 4.3. Feature Importance in RFc

Table 5 shows the feature importance values of FF4 calculated by using RFc (5k), with the values sorted in descending order. Head pose features turned out to be the most important features, followed by most of the AU intensity features (AU45 = blinking, AU20 = horizontal lip stretching, AU23 = tightening of lips, AU17 = chin raising, AU02 = outer brow raising, and AU15 = lip corner depressing), and finally the AU presence features. The results indicate that head pose and intensity features are more informative for behavioural prediction than the presence features are. We believe that the head pose might be particularly informative because it can show the subjects lack of engagement in the conversation. Nevertheless, more data would be necessary for a detailed investigation.

### 4.4. Content Analysis

We analysed the content of the verbal conversation by first transcribing the spoken language into texts and subsequently classifying the texts using binary content parameters, e.g., whether a specific topic [3] was raised (1) or not (0), and meta parameters of the conversations, e.g., individual and group word counts. The obtained variables were analysed with respect to the contribution behaviour and the collected demographic information using a Tobit regression. Table 6 illustrates that the variable “End-Games” has the strongest effect on the group level contributions. As is described in greater detail and discussed with respect to other parameters in [40], the variable codes whether the individuals mentioned the end games specifically. Briefly discussing the idea of making high contributions until the very end has a significant, positive effect on the contributions. This result is, furthermore, possibly in line with the obtained observation that the last parts of FFC are more informative. The conclusion that providing high contributions to the public good until the very end constitutes the final element of the arrangement in the group (for example, 1. What is the optimal contribution level? 2. Does everyone agree on the discussed scheme? 3. Identify the special trait of the last period). This is further supported by the observation that groups that discussed longer in terms of the word counts were on average more successful. However, this effect is stronger when the groups communicate for the first time and weaker when they repeat the communication in the next round. This may be a result of a learning process [40].

## 5. Discussion

In this paper, we proposed two approaches (a facial expression analysis approach and a content analysis approach) to investigate what influences subjects’ contributions in a public goods game utilising a dataset of 127 groups, each group with 4 participants. We automatically analyse the facial expressions of the FFC videos in three steps. First, we extract facial features from each individual face and frame by using OpenFace [47] (see Section 2.1). Second, we calculate the facial activity descriptors (FADs) for each individual face from all the frames (see Section 2.2). Third, we concatenate the activity descriptors of the four individuals in all 4! = 24 possible ways to form our newly proposed group activity descriptors (GADs) (see Section 2.3). All the GADs receive the same label. We classify all 127 videos × 24 permutations = 3048 GADs individually and average the classification outcome of each video to obtain the prediction score (see Section 2.4).

Several experiments in facial expression analysis of FFC were conducted, trying to predict the full-contribution behaviour better than uninformed and informed guessing. The results show that predicting the contribution behaviour of the group based on the facial visual cues communication is challenging; the results were better than uniformed guesses and slightly better than informed guesses (trivial). We expected this task to be particularly difficult because we are trying to predict decisions that have far more hidden influences. The initial results presented are promising and we believe that a larger dataset is necessary to predict cooperation behaviour more reliably. This hypothesis is supported by the results in Figure 2b. The accuracies of high capacity classifiers (RFc with 1k and 5k trees) improve when only a subset of the features (FF4) is used, i.e., the prediction becomes more reliable although less information is available. This behaviour is related to the curse of dimensionality and Hughes phenomenon [53] and is typical for datasets that are too small. Increasing dataset size would offer great potential for more reliable classification in higher dimensional feature spaces and for exploring fine-grained behavioural patterns that help sustain cooperation and avoid free riding in public goods games. In turn, this would find application in public tasks, e.g., policies for the solution of environmental problems or managerial analyses concerning the performance of teams in companies.

We initially assumed that the beginning of the FFC video was more informative when it came to predicting human behaviour, perhaps even just the first seconds. Our findings did not support this assumption, with both the facial expression analysis and content analysis approaches indicating that the last part of the FFC video was probably more informative. However, we were not able to prove this, probably due to an issue of the dataset size.

We looked at the correctly classified and wrongly classified FFC video and found little difference in the behaviour between groups at the beginning of the FFC video. However, groups that do not contribute fully show less commitment later in the FFC video. Therefore, we conclude that the last part of the FFC video can be used to predict the contribution behaviour of groups more reliably because it is easier to establish whether the group is communicating well when the introductory phase has already ended. Furthermore, participants might control their facial expressions more at the beginning of communication, while their spontaneous facial expressions become visible at the end of the communication. Thus, facial expressions and head movements provide valuable information that contributes to the classification of behaviour. In short, the current results show how it is possible to predict whether all the participants in a group will contribute fully in the last period of the public goods game. Although facial expression analysis and content analysis show that the final part of FFC videos seems to be more advantageous for predicting the contribution behaviour of groups, we have no proof that the last part of FFC videos are better than the beginning of the FFC videos due to a lack of data. It is essential to use larger datasets to obtain a deeper understanding of the problem. This would allow cooperation behaviour to be predicted more reliably and our findings to be confirmed.

## Figures and Tables

**Figure 1 sensors-19-02786-f001:**
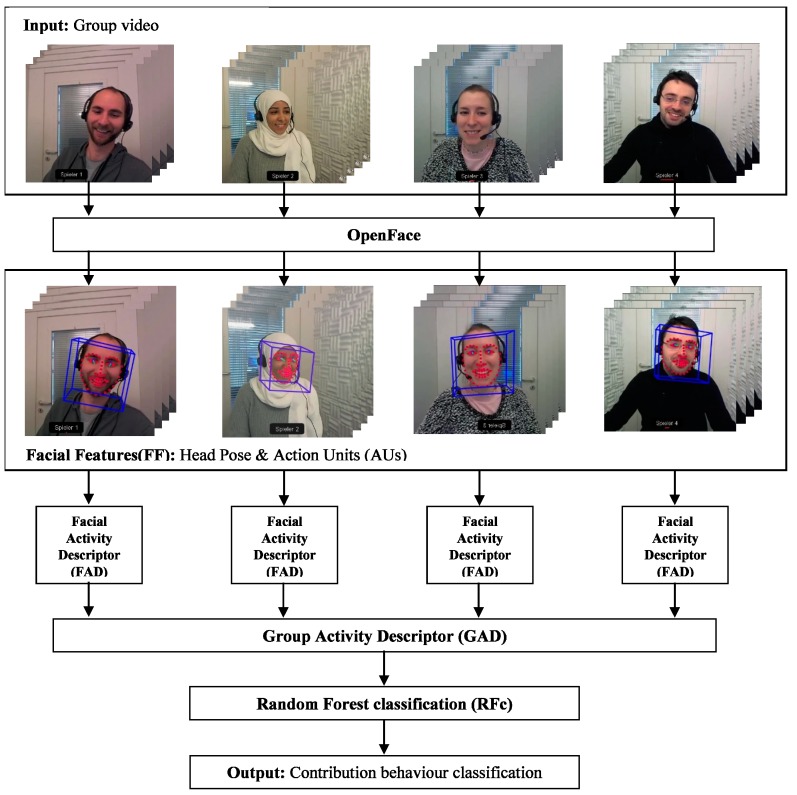
The pipeline of automated facial expression analysis from FFC videos. Each input group video consists of 4 participants communicating for three minutes (in total we have 127 different groups). Each face is first analysed using OpenFace. OpenFace detects the face, facial landmarks, head pose, and extracts facial action units (AUs). AUs and head pose form the facial features for each participant. From these, also for each participant, we calculate facial activity descriptors (FADs). All FADs from all 4 participants are then combined to form the group activity descriptor (GAD). Finally, random forest (RFc) is used to classify the facial contribution behaviour (predicting whether the group will contribute fully or not).

**Figure 2 sensors-19-02786-f002:**
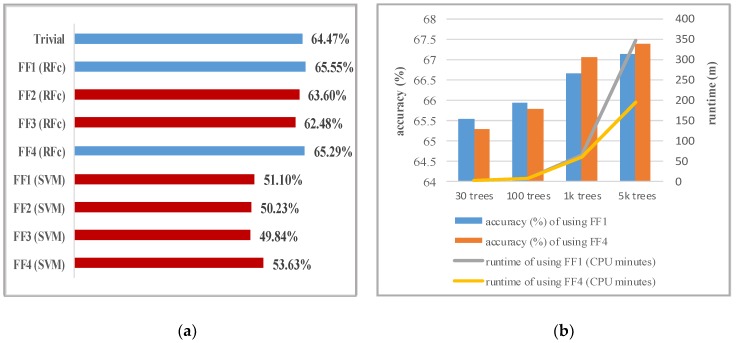
The accuracy of RFc and SVM: (**a**) The accuracy of RFc (30 trees), SVM, and informed guess (trivial) using leave-one-session-out crossvalidation; (**b**) The accuracy and runtime of RFc using FF1 and FF4 with varying number of trees.

**Figure 3 sensors-19-02786-f003:**
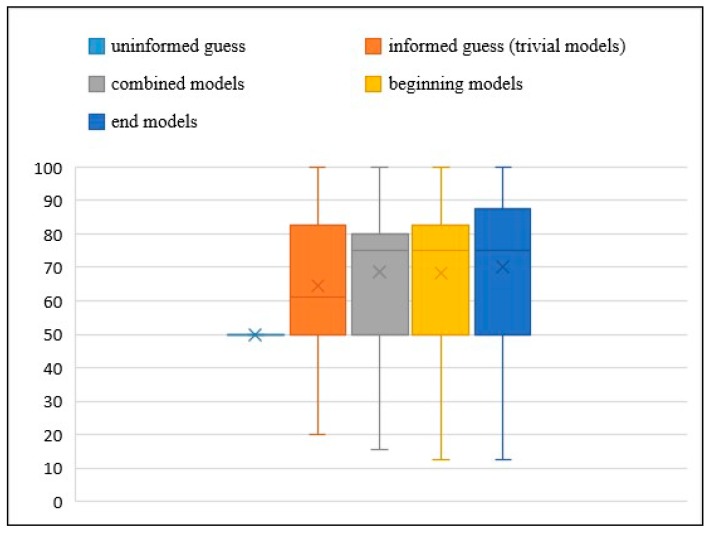
Boxplot of comparing uninformed guess and the accuracy of trivial models with three different RFc (5k trees) models that use FF4 and optimal threshold (combined, beginning and end models). The accuracy of all the RFc models is better than the uninformed guess and the trivial models, with end models obtaining the best accuracy, followed by combined models and beginning models. The crosses represent mean values, the boxes show 25% and 75% quantiles and the median, while the whiskers show the minimum and maximum values.

**Table 1 sensors-19-02786-t001:** List of extracted facial features in OpenFace (I—intensity, P—presence).

Head Pose	AU	AU Full Name	Prediction		AU	AU Full Name	Prediction	
Yaw	AU1	Inner brow raiser	I	P*	AU14	Dimpler	I*	P
Pitch	AU2	Outer brow raiser	I*	P*	AU15	Lip corner depressor	I*	P
Roll	AU4	Brow lowerer	I	P*	AU17	Chin raiser	I*	P
	AU5	Upper lid raiser	I	P*	AU20	Lip stretched	I*	P*
	AU6	Cheek raiser	I	P	AU23	Lip tightener	I*	P*
	AU7	Lid tightener	I*	P	AU26	Jaw drop	I	P*
	AU9	Nose wrinkler	I	P*	AU28	Lip suck	-	P*
	AU10	Upper lip raiser	I	P	AU45	Blink	I*	P*
	AU12	Lip corner puller	I	P				

*Refers to the selected facial features that OpenFace can estimate robustly based on the results of [47].

**Table 2 sensors-19-02786-t002:** The three categories of splits (combined models, beginning and end models) to investigate which categories are most useful. Each split is processed individually by RFc (5k trees) using FF4 and optimal threshold.

Combined Models (11 Splits)		Beginning Models (11 Splits)		End Models (11 Splits)	
1st & 3rd third split	1st & 4th & 5th fifth split	1st third split	3rd fifth split	3rd third split	5th fifth split
1st & 4th quarter split	1st & 2nd & 4th & 5th fifth split	1st quarter split	1st & 2nd fifth split	3rd quarter split	3rd & 4th fifth split
1st & 3rd & 4th quarter split	1st & 2nd & 3rd & 5th fifth split	2nd quarter split	1st & 3rd fifth split	4th quarter split	3rd & 5th fifth split
1st & 2nd & 4th quarter split	1st & 3rd & 4th & 5th fifth split	1st & 2nd quarter split	2nd & 3rd fifth split	3rd & 4th quarter split	4th & 5th fifth split
1st & 5th fifth split	1st & 2nd & 3rd & 4th & 5th fifth split	1st fifth split	1st & 2nd & 3rd fifth split	3rd fifth split	3rd & 4th & 5th fifth split
1st & 2nd & 5th fifth split		2nd fifth split		4th fifth split	

**Table 3 sensors-19-02786-t003:** Paired U-test for comparing the results of the uniformed guess, trivial, combined, beginning and end models using RFc (5k) with FF4 and optimal threshold.

Models	p-Value
uninformed guess & trivial models	0.0000*
uninformed guess & combined models	0.0000*
uninformed guess & beginning models	0.0000*
uninformed guess & end models	0.0000*
trivial models & combined models	0.0530
trivial models & beginning models	0.0519
trivial models & end models	0.0082*
combined models & beginning models	0.9872
combined models & end models	0.4435
beginning models & end models	0.5147

* p < 0.05 (significant difference).

**Table 4 sensors-19-02786-t004:** An overview of the ablation studies of the proposed solutions, with each experiment using RFc with FF4. We conducted 5 experiments combining different methods (ticks display its usage) to improve the model’s performance. The results are the average accuracy of models obtained from cross-validation.

Method	Exp. 1	Exp. 2	Exp. 3	Exp. 4	Exp. 5
Using all frames with RFc	√	√	√		
Using RFc with 5k trees		√	√	√	√
Applying optimal threshold			√	√	√
Using combined models				√	
Using end models (11 splits)					√
Average Accuracy	65.29%	67.39%	68.11%	68.57%	70.18%

**Table 5 sensors-19-02786-t005:** Features importance for the FF4 by using RFc (5k), I – intensity, P – presence. The features in table are ordered by their importance from most important to least important.

Imp	Feature	Value	Imp	Feature	Value	Imp	Feature	Value	Imp	Feature	Value
1	pose_y	0.1387	7	AU17_I	0.1029	13	AU07_I	0.0759	19	AU26_P	0.0696
2	pose_p	0.1367	8	AU02_I	0.1022	14	AU20_P	0.0744	20	AU09_P	0.0667
3	pose_r	0.1349	9	AU15_I	0.1012	15	AU45_P	0.0736	21	AU01_P	0.0640
4	AU45_I	0.1060	10	AU23_P	0.0769	16	AU02_P	0.0721			
5	AU20_I	0.1036	11	AU05_P	0.0766	17	AU28_P	0.0717			
6	AU23_I	0.1033	12	AU14_I	0.0763	18	AU04_P	0.0707			

**Table 6 sensors-19-02786-t006:** Results of Tobit regression of group contributions with p-values denoted in brackets.

Dependent Variable: Group Contributions	Coefficients of Joint Observations	Dependent Variable: Group Contributions	Coefficients of Joint Observations
Number of words	0.088 (0.102)	Number of males	−4.973 (0.361)
End-Games	39.855 (0.006)	Aggregated age	0.108 (0.887)
Invest All	64.899 (0.073)	Constant	3.395 (0.964)
Subjects Against	−23.850 (0.205)	R-square	0.0447
Previous Experience	6.795 (0.554)	Number of Observations	127
Threats & Consequences	−7.159 (0.604)	LR-Chi2	23.01
Number of economists	1.349 (0.819)

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
