# Peer review of "Predicting Group Contribution Behaviour in a Public Goods Game from Face-to-Face Communication"

_sensors, 2019, doi:10.3390/s19122786_

Round 1
Reviewer 1 Report
The authors have addressed my previous concerns. I would like to accept this paper for publication.
Author Response
Response to Reviewers Comments
Dear reviewers,
We sincerely thank you for the constructive comments, which were of great help in improving the paper.
Thank you for your very valuable suggestions.
Note: The manuscript has reviewed and revised by a native speaker.
Best Regards,
Ehsan Othman
(corresponding author)
Reviewer 2 Report
The paper proposes a method to use facial expression analysis and content analysis for describing a group activity within the context of public goods games. The method is based on combining several techniques that are already existing.
While the paper is generally well written, it can be improved.
The introduction section is a bit too long and merged with related work. Consider separating them, and provide more background information and references for the context of the application. The discussion on research motivations and current gaps should be strengthened.
The method section should be before the dataset description. Figure 1 should be improved: only showing 4 subjects give the impression that the system would only be trained to work for these four subjects, and the processes do not show outputs and inputs. fAfter reading this section, readers should have a clearer understanding of public goods games and how it can be possibly described using video, in terms of facial expression and contents.
Experimental section should separate the results and discussion. It should start with the experiment setting, followed by results and explanation on how to interpret the data and why they are interesting. Then finally provide discussions on the interesting findings from the results relative to the hypotheses, leading to the conclusions section.
Conclusion can be much briefer moving much of the details to a separate discussion section (as per previous suggestion), and provide clearer justifications on what have not been resolved by the paper, leading to some suggestions for future work.
Author Response
Response to Reviewers Comments
Dear reviewers,
We sincerely thank you for the constructive comments, which were of great help in improving the paper. In the following, we cite your comments and respond to them, clarifying the changes we made in the revision.
Comment 1: The introduction section is a bit too long and merged with related work. Consider separating them, and provide more background information and references for the context of the application. The discussion on research motivations and current gaps should be strengthened.
Response 1:
a. We separated the related work from the introduction. The related work is now a subsection in the introduction section.
b. We provided more background information and references for the context of the application (see the highlighted paragraph in the related work subsection).
c. We added more clarification about the research motivation and current gap (see the highlighted paragraph in the contribution subsection).
Comment 2: The method section should be before the dataset description. Figure 1 should be improved: only showing 4 subjects give the impression that the system would only be trained to work for these four subjects, and the processes do not show outputs and inputs. After reading this section, readers should have a clearer understanding of public goods games and how it can be possibly described using video, in terms of facial expression and contents.
Response 2:
a. We moved the dataset section after the method section.
b. We improved Figure 1 to make it clearer and understandable.
c. We have explained the public goods game and how it can be possibly described using video, in terms of facial expression and contents (see the highlighted paragraph in the method section).
Comment 3: Experimental section should separate the results and discussion. It should start with the experiment setting, followed by results and explanation on how to interpret the data and why they are interesting. Then finally provide discussions on the interesting findings from the results relative to the hypotheses, leading to the conclusions section.
Response 3:
a. We tried to separate the results section from experimental setting section but discovered the difficulty of tracking the several different experiments with the respective results. Further, by separating them, the results section contained more repetition because of the use of sentences or phrases similar to the experimental section to link each result with its experiment. In our opinion, this change negatively affects readability and clarity of the manuscript. Therefore, we kept them merged.
b. In our last draft, we mistakenly named the discussion section “conclusions”. Thanks for pointing this out.
Comment 4: Conclusion can be much briefer moving much of the details to a separate discussion section (as per previous suggestion), and provide clearer justifications on what have not been resolved by the paper, leading to some suggestions for future work.
Response 4:
a. See response 3b.
b. We added explanations on what has not been resolved by the paper and our suggestion for future work (see the highlighted paragraph in the discussions section).
Thank you for your very valuable suggestions.
Note: The manuscript has reviewed and revised by a native speaker.
Best Regards,
Ehsan Othman
(corresponding author)
This manuscript is a resubmission of an earlier submission. The following is a list of the peer review reports and author responses from that submission.
Round 1
Reviewer 1 Report
1. The title of this paper is too long and redundant. Please make it clear that what is done in this paper. 2. The contribution of this paper is limited. Facial expression recognition has already been widely studied. However, there is no comparison with the state-of-the-art methods. This is the main concern. 3. There are some language issues. For example, in Abstract, "automatic facial expressions analysis" -> "automatic facial expression analysis". A careful proofreading is recommended. 4. The related works are too old. The authors should focus more on recently published papers.
Reviewer 2 Report
The work is interesting by itself, however there is no comparison with any state-of-the-art technique. Most importantly given the setup is subject to multimodal expression analysis, there is no work/discussion has been done. In line with that works such as:
Noroozi, Fatemeh, et al. "Audio-visual emotion recognition in video clips." IEEE Transactions on Affective Computing (2017).
need to be cited and COMPARED with the proposed technique. Additionally recent works such as:
Grobova, Jelena, et al. "Automatic hidden sadness detection using micro-expressions." Automatic Face & Gesture Recognition (FG 2017), 2017 12th IEEE International Conference on. IEEE, 2017.
Kulkarni, Kaustubh, et al. "Automatic recognition of facial displays of unfelt emotions." IEEE Transactions on Affective Computing (2018).
Guo, Jianzhu, et al. "Dominant and Complementary Emotion Recognition From Still Images of Faces." IEEE Access 6 (2018): 26391-26403.
need to be cited and COMPARED with the proposed technique.